# Concordance between the In Vivo Content of Neurospecific Proteins (BDNF, NSE, VILIP-1, S100B) in the Hippocampus and Blood in Patients with Epilepsy

**DOI:** 10.3390/ijms25010502

**Published:** 2023-12-29

**Authors:** Maria A. Tikhonova, Anna A. Shvaikovskaya, Svetlana Y. Zhanaeva, Galina I. Moysak, Anna A. Akopyan, Jamil A. Rzaev, Konstantin V. Danilenko, Lyubomir I. Aftanas

**Affiliations:** 1Scientific Research Institute of Neurosciences and Medicine (SRINM), 630117 Novosibirsk, Russiazhanaevasy@neuronm.ru (S.Y.Z.); liaftanas@neuronm.ru (L.I.A.); 2FSBI “Federal Center for Neurosurgery”, 630087 Novosibirsk, Russiad_rzaev@neuronsk.ru (J.A.R.)

**Keywords:** human, brain, serum, peripheral mononuclear blood cells, hippocampal sclerosis, mesial temporal lobe epilepsy, neurosurgery, immunoassay, immunohistochemical analysis

## Abstract

The identification of reliable brain-specific biomarkers in periphery contributes to better understanding of normal neurophysiology and neuropsychiatric diseases. The neurospecific proteins BDNF, NSE, VILIP-1, and S100B play an important role in the pathogenesis of neuropsychiatric disorders, including epilepsy. This study aimed to assess the correspondence of the expression of BDNF, NSE, VILIP-1, and S100B in the blood (serum and peripheral blood mononuclear cells (PBMCs)) to the in vivo hippocampal levels of subjects with drug-resistant epilepsy who underwent neurosurgery (N = 44) using multiplex solid-phase analysis, ELISA, and immunohistochemical methods, as well as to analyze the correlations and associations of the blood and hippocampal levels of these proteins with clinical parameters. We first studied the concordance between in vivo brain and blood levels of BDNF, NSE, VILIP-1, and S100B in epileptic patients. A positive correlation for NSE between hippocampal and PBMC levels was revealed. NSE levels in PBMCs were also significantly correlated with average seizure duration. BDNF levels in PBMCs were associated with seizure frequency and hippocampal sclerosis. Thus, NSE and BDNF levels in PBMCs may have potential as clinically significant biomarkers. Significant correlations between the levels of the neurospecific proteins studied herein suggest interactions between BDNF, NSE, VILIP-1, and S100B in the pathophysiology of epilepsy.

## 1. Introduction

The identification of reliable biomarkers is a key approach to the early diagnosis of pathological processes in organisms. The state of a human brain is usually evaluated by analyzing brain-specific substances in peripheral biological fluids that are easily accessible for research, i.e., blood; urine; saliva; and, less often, cerebrospinal fluid (CSF). However, due to the blood–brain barrier (BBB) and a number of other factors, such correlations or associations may be weak or completely absent. The concentration of neurospecific proteins in the blood is considered a potential marker of the development and dynamics of neuropsychiatric diseases, as well as response to therapy. In this regard, the question of particular importance is whether there is concordance between the concentrations of these substances in the central nervous system (CNS) and those in peripheral tissues in humans, as in some animals, as well as whether it is possible to use the concentration in periphery to obtain reliable information about its content in the brain. Gaining knowledge of the extent to which measurements of neurospecific substances in the periphery can predict brain health is important from both scientific and practical points of view. Neurospecific proteins, in particular BDNF, NSE, VILIP-1, and S100B, play an important role in the pathogenesis of neuropsychiatric diseases. Therefore, the diagnostic and prognostic value of their content is of significant interest.

Brain-derived neurotrophic factor (BDNF) is a small protein produced in various organs and systems, but it plays a special role in the CNS, enabling the processes of neurogenesis and neuroplasticity. BDNF is perhaps the best indicator of brain neuroplasticity among all other biochemical parameters. Disturbances in the synthesis, processing, or transport of BDNF can lead to various neurological diseases and mental disorders, including Alzheimer’s disease, Huntington’s chorea, Rett syndrome, schizophrenia, depression, epilepsy, etc. [1]. BDNF does not cross the BBB well, so its concentration in the blood may not correlate with its concentration in the brain [2]. Circulating BDNF is found in serum and plasma, and large amounts are stored in human platelets [3] as well as in leukocytes, in which its levels may better correspond to those in brain neurons than serum levels do [4]. An increase in BDNF-TrkB signaling in lymphocytes was shown in patients with Parkinson’s disease after intensive rehabilitation [5]. An increase in BDNF-TrkB signaling was also observed both in lymphocytes and in the brains of rats after repeated transcranial magnetic stimulation [6]. Unlike blood, CSF is considered to be poorly suitable for the determination of BDNF, presumably due to the low analyte content and the invasiveness of the method [7].

Our review of studies on animals revealed varying degrees of concordance between the contents of BDNF in the blood and brain (on average, the strength of the correlations was moderate; [8]). In humans, there are a few studies comparing BDNF levels in the brain and peripheral tissues. In a systematic review of the literature, we found only three studies, in which no significant correlations between the levels of BDNF in the brain and in human plasma/serum were reported [9]. However, it should be noted that two of these studies were performed using post mortem material, and the in vivo study had a very small sample size (nine children with epilepsy), and no analysis of the correlations was reported. Thus, the concordance between in vivo BDNF levels in the brain and those in peripheral substrates, as well as the diagnostic and clinical significance of the latter in various diseases of the nervous system, remain unclear and require additional research.

In vivo detection of an analyte content in the human brain is a non-trivial and difficult task due to the limited availability of human brain tissue for research because of ethical standards and technical restrictions. One available approach is to take brain tissue during brain surgery [9]. This work was performed on biosamples of hippocampal tissue and blood taken in vivo from patients who underwent elective hippocampal resection surgery for drug-resistant epilepsy.

Since the biosamples from patients with epilepsy were studied, in addition to BDNF, the levels of neurospecific proteins (neuron-specific enolase (NSE), visinin-like protein 1 (VILIP-1), and S100 calcium-binding protein B (S100B)) considered to be biomarkers of epilepsy were also determined. It should be noted that NSE is the most widely studied and used biochemical marker for assessing neuronal damage. It can be measured in both serum and CSF. NSE is considered a highly specific marker of damage to the peripheral or central nervous system and a prognostic indicator in various disorders, including epilepsy [10,11]. Serum and CSF NSE levels elevate during the first 48 h after a seizure and correlate with the outcome and duration of epilepsy [12,13]. NSE has also been found in red blood cells and platelets [14,15], and minimal hemolysis can lead to increased NSE in the blood, which may affect the accuracy and clinical value of serum NSE as biomarker [16].

In this regard, researchers are searching for other serological biomarkers of neuronal damage in epilepsy with high sensitivity and specificity. Recently, VILIP-1 and S100B have been proposed as biomarkers with greater specificity for diagnosing epilepsy and determining the severity of the disease. VILIP-1 is a neuron-specific calcium-binding sensor protein abundantly expressed in the CNS [17] which enters the CSF after brain cell destruction [18,19]. It was initially studied as a biomarker in stroke patients and was identified as a useful marker of neuronal damage due to stroke, Alzheimer’s disease, and traumatic brain injury [20]. Previous studies have confirmed that during the BBB dysfunction caused by epilepsy, specific brain proteins can enter the peripheral blood [21]. In a study by Tan et al. (2020) [22], serum VILIP-1 levels increased in patients after epileptic seizure compared to control subjects. Moreover, the accuracy and utility of serum VILIP-1 were higher than those of serum NSE, indicating that serum VILIP-1 may be a better biomarker for assessing neuronal damage caused by seizures, brain injuries, and other pathological conditions. The S100 protein is also a good biomarker for epilepsy. S100B is a Ca^2+^-binding protein that is expressed mainly by astrocytes. Serum S100B levels have been found to be significantly elevated in patients with epilepsy or infants with epileptic seizures, and S100B has been proposed by many authors as a clinically relevant peripheral biomarker in epilepsy [23,24,25,26]. Thus, increases in the concentration of NSE, VILIP-1, or S100B in the blood serum are regarded as indicators of neuronal death during epileptic seizures as well as of severity of the disease.

BDNF expression with an epileptogenic focus in the brain has not been studied yet with regard to changes in its content in the blood. There are no data on the correlation between BDNF and the neurospecific proteins NSE, VILIP-1, and S100B, which reflect brain damage in epilepsy. The study was aimed at assessing the correspondence of the expression of BDNF, NSE, VILIP-1, and S100B in the blood (serum and leukocytes) to the in vivo levels in the hippocampus in patients with drug-resistant mesial temporal lobe epilepsy (MTLE), as well as to analyze the correlations and associations of the expression of these neurospecific proteins in the blood and hippocampus with clinical parameters (i.e., gender, age, duration of the disease, seizure duration and frequency, surgical outcomes, and anti-epileptic drug therapy before surgery).

## 2. Results

The results regarding the expression of BDNF, NSE, VILIP-1, and S100B in the hippocampus and blood are summarized in Table 1. Notably, according to IHC analysis, we revealed moderate significant correlations between the hippocampal levels of BDNF and NSE (*rho* = 0.51, *p* = 0.0006, N = 42) and PBMC levels of BDNF and S100B (*rho* = −0.49, *p* = 0.0065, N = 29). According to a biochemical analysis, moderate significant correlations between VILIP-1 and S100B levels were found in the hippocampus (*rho* = 0.34, *p* < 0.05, N = 39), in PBMCs (*rho* = 0.39, *p* < 0.05, N = 41), and in serum (*rho* = 0.39, *p* < 0.01, N = 41).

### 2.1. BDNF

The results regarding the concordance of BDNF levels in the hippocampus and blood, as well as an analysis of clinical correlates of brain or blood BDNF levels, are summarized in Table 2.

According to a biochemical analysis, BDNF levels in the hippocampus did not correlate with its levels in PBMCs or blood serum. Significant, positive correlations of BDNF levels in PBMCs with an average seizure frequency per month (*rho* = 0.46, *p* < 0.01; Figure 1) and between BDNF levels in serum and the ages of patients (*r* = 0.32, *p* < 0.05) were revealed. A tendency of increased BDNF levels in PBMCs was found in patients with hippocampal sclerosis compared to patients with hippocampal gliosis (*p =* 0.11). BDNF levels in the serum of patients who were treated with drugs regulating ion channel activity were 1.4 times lower than those who were not (*p* < 0.05).

According to IHC analysis, the BDNF levels in the hippocampus did not correlate with the levels in PBMCs. No significant correlations or associations of BDNF levels in the hippocampus or blood with age, sex, disease duration, seizure frequency, average seizure duration, surgery outcome, or anti-epileptic drug therapy before surgery were revealed. At the same time, the BDNF levels in PBMCs of patients with hippocampal sclerosis were markedly higher than those of patients with hippocampal gliosis (*p* < 0.01; Figure 2). A similar tendency to increase was found in the hippocampus.

### 2.2. NSE

The results regarding the concordance of NSE levels in the hippocampus and blood, as well as those of our analysis of clinical correlates of brain and blood NSE levels, are summarized in Table 3.

According to biochemical analysis, there was a moderate positive correlation between NSE levels in the hippocampus and its levels in PBMCs (*rho* = 0.55, *p* < 0.001; Figure 3). No significant correlations or associations between NSE levels in the hippocampus or blood (PBMCs and serum) and age, sex, disease duration, seizure frequency, or surgery outcome were found. At the same time, drugs inhibiting the activity of NMDA receptors influenced NSE concentrations in the hippocampus and serum. The NSE levels in the hippocampuses of patients who were treated with those drugs were lower (*p* < 0.05), while in serum, they were higher (*p* < 0.05) than in patients who were not.

According to IHC analysis, there was a tendency of correlation between NSE levels in the hippocampus and its levels in PBMCs (*rho* = 0.3, *p* = 0.057). No significant correlations or associations between NSE levels in the hippocampus or blood and age, sex, disease duration, seizure frequency, surgery outcome, or anti-epileptic drug therapy before surgery were revealed. At the same time, the NSE levels in PBMCs were significantly correlated with the average seizure duration (*rho* = 0.34, *p* < 0.05; Figure 4).

### 2.3. S100B

The results regarding the concordance of S100B levels in the hippocampus and blood, as well as those of our analysis of the clinical correlates of brain or blood S100B levels, are summarized in Table 4.

According to our biochemical analysis, there were no significant correlations of S100B levels in the hippocampus with its levels in PBMCs or serum. S100B levels in serum were positively correlated with the age of the patient (*rho* = 0.34, *p* < 0.05). No other significant correlations or associations were identified.

According to IHC analysis, the S100B levels in the hippocampus did not correlate with the levels in PBMCs. No significant correlations or associations between S100B levels in the hippocampus or blood and age, sex, disease duration, seizure frequency, average seizure duration, surgery outcome, or anti-epileptic drug therapy before surgery were revealed.

### 2.4. VILIP-1

The results regarding the concordance of VILIP-1 levels in the hippocampus and blood, as well as our analysis of the clinical correlates of brain or blood VILIP-1 levels, are summarized in Table 5.

According to the biochemical analysis, VILIP-1 levels in the hippocampus did not correlate with its levels in PBMCs or serum. No significant correlations or associations between VILIP-1 levels in the hippocampus or blood and age, sex, disease duration, seizure frequency, average seizure duration, surgery outcome, or anti-epileptic drug therapy before surgery were revealed.

## 3. Discussion

The identification of reliable brain-specific biomarkers in periphery contributes to a better understanding of normal neurophysiology, as well as the pathological processes related to neuropsychiatric diseases and their early diagnosis, therapy response, or prognosis. Hence, the study of neurospecific molecules and the correspondence of their content between the brain and periphery is of great interest. However, quite a few studies have investigated the correlation for neurospecific molecules between the brain and periphery in humans [9]. For example, the study by Gadad et al. (2021) revealed a positive correlation between the levels of IL-6 and GDNF in the brain (Brodmann area) and blood plasma [27].

Analytes in the human brain can be measured post mortem or in vivo, either following brain surgery or using neuroimaging techniques. It should be noted that, compared to in vivo studies, the validity of the post mortem brain measurements may be compromised by proteolytic degradation of some molecules with the time elapsed from death to sample conservation [28]; by the post mortem stability of some proteins and the susceptibility to post mortem degradation of the others [29]; or by significant leakage of brain-specific substances into the blood due to dramatically compromised permeability of the BBB at the terminal phase of a disease and after death. Thus, in vivo human studies are of particular value, although in vivo determination of analyte content in the human brain is a non-trivial and difficult task due to the limited availability of human brain tissue for research because of ethical standards and technical capabilities. Further progress in the field would be provided by the development of advanced in vivo metabolomic analyses, neuroimaging techniques, and blood assays for exosomes of brain origin, as well as the emergence of novel approaches [9].

Among human peripheral tissues and fluids, the blood attracts particular attention because it is easily accessible for research and wide screening in populations (vs. CSF or nasal epithelium). On the other hand, the concentration of neurospecific proteins in the blood serum is considered to be closer to the brain content than in the urine or saliva. Moreover, PBMCs express many CNS enzymes, receptors, and downstream signaling proteins, and are considered to be a suitable cellular model for drug discovery and studying the mechanisms of neuropsychiatric disorders [30]. Here, we studied the correspondence of the expression of BDNF, NSE, VILIP-1, and S100B in the blood (serum and PBMCs) with the in vivo content in the hippocampuses of patients with drug-resistant epilepsy using biosamples obtained during neurosurgical operations. We also analyzed the correlations and associations of the expression levels of these neurospecific proteins in the blood and hippocampus with clinical parameters to estimate the clinical value of the biomarkers.

Serum BDNF levels are low (e.g., 100–1300 pg/mL according to [31]), and vary between individuals (10–50-fold difference) as well as during the day (3–4 times) [31]. Due to high variability, standards for the content of BDNF in the blood have not been developed. In epilepsy, the BDNF content may be elevated in the hippocampus. BDNF produced by hippocampal cells can be transported along axons (mossy fibers) and reach their terminals in the CA regions of the hippocampus [32]. Abnormal growth of mossy fibers into the dentate gyrus in the CA regions leads to the formation of excitatory synapses, creating electrical circuits with the ability to synchronize and generate epileptic seizures. Often, such epileptic seizures are pharmacoresistant and can be stopped only by resection of the epileptogenic focus. The initiation of epileptogenesis after brain injury is associated with increased levels of BDNF and other neurotrophins in the hippocampus [33]. BDNF may exert a proepileptic role by potentiating excitatory synapses for epileptogenesis. Therefore, the frequency and severity of seizures, especially in chronically damaged hippocampuses, could be regulated by changes in BDNF content. However, compared with groups of healthy individuals, both increased [34] and decreased [35] BDNF levels were found in the blood of patients with epilepsy. The average level of BDNF in serum that we detected herein (median = 3458.7 pg/mL) indicates an increased level of the analyte in the blood of patients with drug-resistant epilepsy. Nevertheless, clinically significant correlations of peripheral BDNF levels were demonstrated earlier. For example, in children with severe traumatic brain injury, correlational links between serum neural marker contents and assessments on the Glasgow scale were revealed for BDNF, and serum BDNF levels were normal in cases with favorable traumatic brain injury outcomes [36].

According to our results, the BDNF levels in the hippocampus did not correlate with its levels in PBMCs or blood serum. These results are in accordance with previous reports on the absence of a correlation of BDNF between the brain and the serum/plasma in two post mortem studies [27,37], as well as with our preliminary results obtained in 20 epileptic patients who underwent brain surgery [38]. At the same time, BDNF levels in the PBMCs of patients with hippocampal sclerosis were markedly higher than those of patients with hippocampal gliosis. A similar tendency to increase was found in the hippocampus. These findings generally agree with our previous findings on negative correlation between hippocampal BDNF levels and the volumes of the different subfields of the hippocampus measured by MRI in patients with drug-resistant MTLE [39]. Moreover, a significant, positive, moderate correlation of BDNF levels in PBMCs with an average seizure frequency per month was revealed. These findings evidence the proepileptic nature of BDNF action. In addition, we found a weak positive correlation between BDNF levels in serum and the age of the patients. Serum BDNF levels differed significantly in patients who were treated with drugs regulating ion channel activity compared to those who were not. However, this result should be interpreted with caution, as the number of patients in the drug-free group was low. Although no concordance between the hippocampal and blood BDNF levels was established, BDNF levels in PBMCs were correlated with seizure frequency and hippocampal sclerosis. Thus, this parameter may have potential as a clinically significant biomarker.

NSE is a glycolytic enzyme involved in the oxidation of glucose during aerobic glycolysis in nervous cells that results in the formation of ATP, a vital molecule for energy supply and the vital functioning of cells. According to modern concepts, the lack of ATP due to disturbed glycolysis in the brain is one of the main causes of epileptic activity [40]. During convulsive conditions in animal studies, the content of NSE increased in some brain structures, but decreased in epileptogenic foci [41]. Disturbances in energy metabolism are closely associated with deficits in the functioning of ion channels and the death of brain cells. When neurons are damaged and the integrity of the BBB is disrupted, NSE is released into the CSF and blood [42]. Quantitative determination of NSE in the blood serum can be used to diagnose neuroendocrine tumors [43], ischemic stroke [44,45,46], and neuroblastoma [47]. There is also evidence of an increase in NSE levels in the blood serum of subjects with seizures [48].

In the present study, serum NSE levels were lower compared to the results of other, similar studies. For example, high levels of NSE in the blood were found in conditions involving severe damage to the BBB (e.g., massive TBI) or in patients after cardiac arrest. In a post mortem study by Cronberg et al. (2011), the severity and extent of ischemic brain damage were positively correlated with NSE concentrations [49]. However, in children with traumatic brain injuries, NSE levels on the first day after the trauma were not a strict criterion for injury outcomes [50]. The low levels of NSE in the serum might be also attributed to differences in the duration of the disease. Our study involved patients with long-term drug-resistant epilepsy, while aforementioned similar studies enrolled patients with single/rare episodes of epileptic seizures [51] or who had recently been diagnosed with epilepsy [52]. Moreover, serum NSE levels did not correlate with NSE levels in the hippocampus, nor were there significant correlations or associations of NSE levels in the serum with age, sex, disease duration, seizure frequency, or surgery outcome. Although the serum and hippocampal NSE levels differed significantly in patients who were treated with drugs inhibiting the activity of NMDA receptors compared to those who were not, this result should be interpreted with caution, as the number of patients in the drug-treated group was low. Nevertheless, we found a moderate, positive correlation of NSE between the hippocampus and leukocytes. NSE levels in PBMCs were also significantly correlated with the average seizure duration. Thus, NSE levels in PBMCs may have potential as a clinically significant biomarker.

There were no significant correlations of S100B or VILIP-1 levels in the hippocampus with their levels in PBMCs or serum. A weak, positive correlation of S100B levels in the serum with the age of the patient was found. No other significant correlations or associations with clinical indices were identified. Noteworthy, there was a moderate positive correlation between the hippocampal levels of BDNF and NSE and a negative correlation between the PBMC levels of BDNF and S100B. Significant positive correlations between VILIP-1 and S100B levels were found in the hippocampus, PBMCs, and serum. Thus, we have not revealed a concordance between the hippocampal and blood levels of VILIP-1 or S100B in epileptic patients, nor have correlations or associations of the blood levels of these analytes with clinically significant parameters been disclosed. Nonetheless, these biomarkers might be taken into account in further studies on epilepsy with a focus on their relation to seizures, which was revealed earlier [22,25], or on their interactions with other neurospecific proteins involved in the pathogenesis of epilepsy (e.g., BDNF).

### Limitations

This study has several limitations. First, we used a relatively small sample size (N = 44). Hence, we focused on moderate or strong correlations and highly significant associations revealed by both methods (biochemical and IHC) which were used. Other significant results are reported with caution as preliminary ones. Second, we recruited only patients with long histories of drug-resistant MTLE, since these are notoriously the prime candidates for surgery. While undoubtedly useful and valuable, caution should be taken in regards to the generalization of these findings, because a concordance between the content of neurospecific proteins in different tissues might be affected by the disease or drug therapy. Additionally, being limited by the sample size, we should take the results regarding drug effects as well as associations with surgery outcomes with caution due to the small number of patients in some groups. Further studies on larger samples and subjects with other pathologies that use advanced statistical tools and intervention techniques are clearly needed.

## 4. Materials and Methods

### 4.1. Patients

The subjects in this study were patients diagnosed with drug-resistant MTLE according to the diagnostic guidelines [53] who were surgically treated at the Federal Center of Neurosurgery from 2021 to 2022. A total of 44 patients (24 males and 20 females, 19–56 (median = 34 (28.5; 39)] years of age) with morphologically proven hippocampal sclerosis or gliosis participated in the study. All patients underwent noninvasive video-EEG monitoring and high-resolution brain MRIs before surgery. All patients underwent anterior temporal lobectomies. Detailed information about the patients is provided in Appendix A. Each patient signed a written informed consent form to participate in the study. The study was conducted in accordance with the Declaration of Helsinki and approved by the local Ethics Committee of the Federal Center for Neurosurgery, Novosibirsk, Russia (protocol No. 1, dated 1 March 2019).

### 4.2. Sampling Procedures

Peripheral blood samples were taken before surgery after overnight fasting. Then, premedication was given, which was the same for all patients (Phentanyl 0.005% (GosZMP, Moscow, Russia); Propofol (Armavir biofactory, Krasnodar, Russia). Hippocampal resection was performed 3.3 ± 0.4 h after the administration of anesthesia. Areas of the basal parts of the temporal lobe up to 5 cm from the pole and amygdala were resected.

Blood was taken in amounts of 4 mL into two tubes with clotting activators for biochemical analysis. The first tube was incubated for 30 min (for NSE and S100B analysis); the second tube for 60 min (for VILIP-1 and BDNF analysis). After centrifugation (2000× *g* for 10 min) and aliquoting, protease inhibitors were added in amounts of 2 μL/mL of serum; then, the serum samples were frozen at −20 °C.

To isolate peripheral blood mononuclear cells (PBMCs), blood was placed into 8 mL tubes with separation gel, incubated at room temperature for 1–1.5 h, and centrifuged at 2500× *g* for 20 min. The layers of plasma with mononuclear leukocytes were transferred into clean tubes, and leukocytes were washed from the plasma three times in phosphate buffer (1xPBS). The suspensions of mononuclear cells were resuspended in 200 μL of the buffer and the leukocyte formula was determined; then, the suspension was transferred into two 75 μL tubes. Protease inhibitors were added to one of them in an amount of 0.375 μL; then, the samples were frozen and stored at −20 °C for biochemical analysis. For immunohistochemical (IHC) analysis, leukocytes were fixed with 4% formaldehyde, washed with 1xPBS, applied to four gelatinized glass slides, dried, and stored at +4C.

Hippocampal samples (~1 cm^3^ in volume) were placed in cooled (+4 °C) 1xPBS. Then, each sample was divided into two parts. One was frozen and stored at −70 °C for biochemical analysis. The second was placed into a container with a fixative (4% paraformaldehyde) for one day, then into a solution of 30% sucrose at 4 °C to dehydrate the samples for 3–4 weeks. After being immersed in the embedding Tissue-Tek O.C.T. compound (Sakura Finetek, Torrance, CA, USA), the samples were frozen and stored at −70 °C until being sectioned into slices 30 μm thick with a cryostat HistoSafe MicroCut—SADV (Citotest Scientific Co.,Ltd., Nanjing, China) for IHC analysis.

### 4.3. Biochemical Analysis

Samples of the hippocampus and PBMCs for biochemical analysis were homogenized with ULTRA-TURRAX homogenizer (IKA-Werke GmbH & Co. KG, Staufen, Germany) in cold homogenizing buffer at a temperature of 4 °C for 30 s, followed by incubation in lysis buffer (50 mM Tris, pH 8.0, 1% NP-40, 0.5% sodium deoxycholate, 0.1 SDS, 2 mM EDTA, and cocktail of inhibitors (P1860, Sigma-Aldrich, Darmstadt, Germany)) for 30 min at 4 °C. Cellular debris was removed by centrifugation at room temperature at 18,000× *g* for 15 min. The supernatants were used to assess BDNF, NSE, S100B, and VILIP-1 levels. Total protein concentrations were estimated using the Bradford assay (Bio-Rad, Hercules, CA, USA).

BDNF levels in serum, hippocampal, and PBMC extracts were measured by multiplex solid-phase analysis using a multiplex analyzer of proteins and nucleic acids (MILLIPLEX Luminex 200, Merck KGaA, Darmstadt, Germany) and xMAP technology with a Multiplex Assays reagent for BDNF (HNDG3MAG-36K-01 Human Neurodegenerative Disease Magnetic Bead Panel 3, Merck KGaA, Darmstadt, Germany). S100B levels were analyzed using enzyme-linked immunosorbent assays (ELISAs) with a DuoSet ELISA kit (DY1820-05; R&D Systems, Minneapolis, MN, USA), according to the manufacturer’s instructions. VILIP-1 levels were measured using ELISA kits (Cloud-Clone Corp., Wuhan, China) according to the manufacturer’s instructions. NSE levels were measured using ELISA kits (Vector-BEST, Novosibirsk, Russia) according to the manufacturer’s instructions. Concentrations of BDNF, VILIP-1, S100B, and NSE in serum were expressed in pg/mL, while in samples from the hippocampus and PBMCs, the concentrations were presented per mg of total protein.

### 4.4. Immunohistochemical (IHC) Analysis

IHC analysis was performed in biosamples of the hippocampus and PBMCs for analytes that were expressed in both types of tissues (BDNF, S100B, and NSE). For rehydration and antigen retrieval, samples were incubated in Trilogy solution (Sigma-Aldrich Co., Darmstadt, Germany) for 1 h. Then, non-specific binding was blocked using 1% bovine serum albumin, followed by overnight incubation of the samples at 4 °C with primary antibodies. After incubation with the respective secondary antibodies, the slices were covered with Fluoromount containing 4′,6-diamidino-2-phenylindole (DAPI; Abcam, Cambridge, UK). Primary antibodies against BDNF (1:200; NB100-98682, Novus Biologicals, Littleton, CO, USA) and secondary goat-anti-rabbit Alexa Flur 488 antibodies (1:600; ab150077, Abcam, Cambridge, UK) were used to evaluate the BDNF expression. Primary antibodies against NSE (1:200; PAA537Hu01, Cloud-Clone Corp., Wuhan, China) and secondary goat-anti-rabbit Alexa Flur 488 antibodies (1:600; ab150077, Abcam, Cambridge, UK) were applied to estimate the NSE expression. Primary antibodies against S100B (1:200; MAA567Hu22, Cloud-Clone Corp., Wuhan, China) and secondary goat-anti-mouse Alexa Fluor 568 antibodies (1:400; ab175473, Abcam, Cambridge, UK) were applied to estimate the S100B expression. Figure 5 shows examples of BDNF, S100B, and NSE immunoreactivity in the hippocampus and PBMCs.

The fluorescence images were finally obtained using an Axioplan 2 (Carl Zeiss, Oberkochen, Germany) imaging microscope and a confocal laser scanning microscope (LSM 780 NLO (Carl Zeiss, Oberkochen, Germany)), and then analyzed via Image-Pro Plus Software 6.0 (Media Cybernetics, Rockville, MD, USA). The fluorescence intensity associated with the expression of BDNF, NSE, or S100B was measured as background-corrected optical density, with the subtraction of staining signals of the non-immunoreactive regions in the images which were converted to grayscale. Negative control samples with the omitted primary antibody emitted only a minimal autofluorescent signal. Four tissue slices per patient and five areas per slice were analyzed, and the mean fluorescence intensity was measured and normalized per area. The areas of interest were 2111 μm^2^ in the hippocampal samples and 106 μm^2^ in PBMCs. For each image acquisition, all imaging parameters were kept the same.

### 4.5. Statistical Analysis

The normality of the data distribution was determined using the Shapiro–Wilk *W* test. The relationship between the measured variables was assessed with the Spearman correlation coefficient (*rho*) in the case of non-normal distribution of the data, or the Pearson correlation coefficient (*r*) for normally distributed data. The correlation between the parameters was termed as weak, moderate, or strong according to correlation coefficients (*r* for Pearson or *rho* for Spearman statistics) of 0.3–0.5, 0.5–0.7, or >0.7, respectively (absolute values).

For the comparison of metric variables grouped by one categorical variable, we applied the Mann–Whitney test and Kruskal–Wallis *H* test, followed by multiple comparisons of mean ranks for all groups in the case of non-normal distribution of the data, or paired Student’s *t*-test and one-way analysis of variance (ANOVA) for normally distributed data.

Data are presented as the median and interquartile range (Q1; Q3) or the mean ± S.E.M. The level of significance was defined as *p* < 0.05 for all experiments reported herein.

## 5. Conclusions

Quite a few studies have investigated the correlation of neurospecific molecules between the brain and periphery (blood) in humans, especially in vivo. Here, we first studied the concordance between in vivo brain (hippocampal) and blood (serum and PBMCs) levels of BDNF, NSE, VILIP-1, and S100B in epileptic patients. We revealed a moderate positive correlation of NSE between the hippocampal and PBMC levels. NSE levels in PBMCs were also significantly correlated with an average seizure duration. There was no concordance between the hippocampal and blood BDNF levels that agreed with previous reports of post mortem studies on subjects with Alzheimer’s disease, those with mood disorders, and healthy controls. Nonetheless, the BDNF levels in PBMCs were correlated with seizure frequency and hippocampal sclerosis. Our findings evidence the proepileptic nature of BDNF action. Thus, NSE and BDNF levels in PBMCs may have potential as clinically significant biomarkers. Notably, there was a moderate positive correlation between hippocampal levels of BDNF and NSE and a negative correlation between PBMC levels of BDNF and S100B. Significant positive correlations between VILIP-1 and S100B levels were found in the hippocampus, PBMCs, and serum. These results suggest interactions between BDNF, NSE, VILIP-1, and S100B in the pathophysiology of epilepsy that should be addressed in future studies.

## Figures and Tables

**Figure 1 ijms-25-00502-f001:**
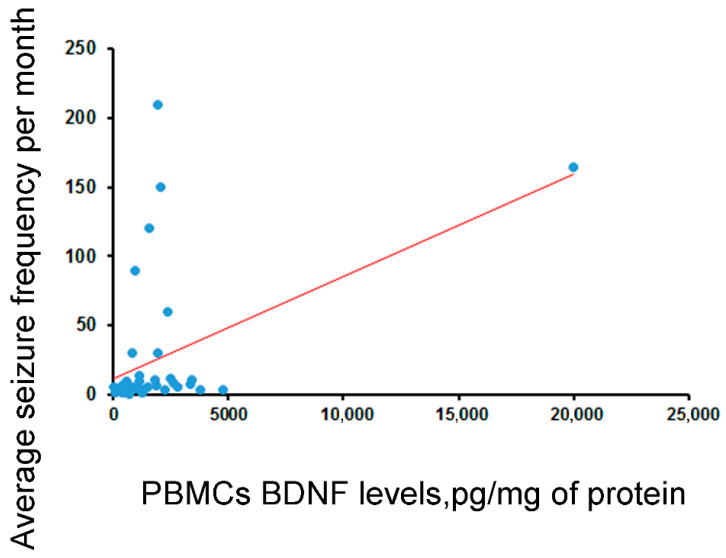
Correlation of concentration of BDNF (corrected by total protein) in PBMCs and average seizure frequency in 40 operated epileptic patients (*rho* = 0.46, *p* < 0.003, Spearman test). The regression line is shown.

**Figure 2 ijms-25-00502-f002:**
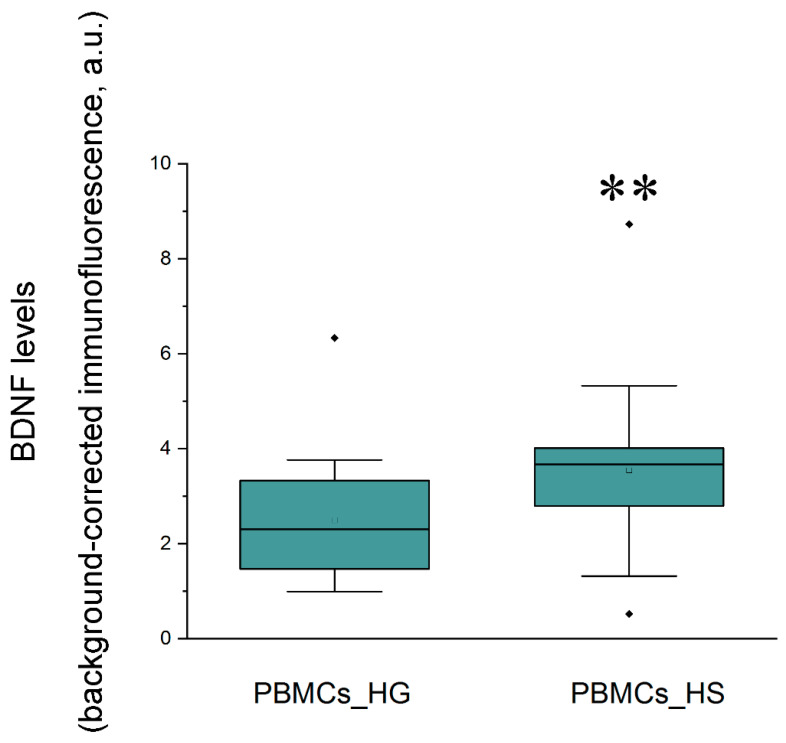
BDNF levels in PBMCs of patients with hippocampal sclerosis (HS) vs. those with hippocampal gliosis (HG). The data are expressed as box-plots with the first (Q1 or 25th percentile) and third (Q3 or 75th percentile) quartiles, median, and whiskers. N = 25 (HS), N = 18 (HG). Statistically significant differences: ** *p* < 0.01 vs. PBMCs_HG.

**Figure 3 ijms-25-00502-f003:**
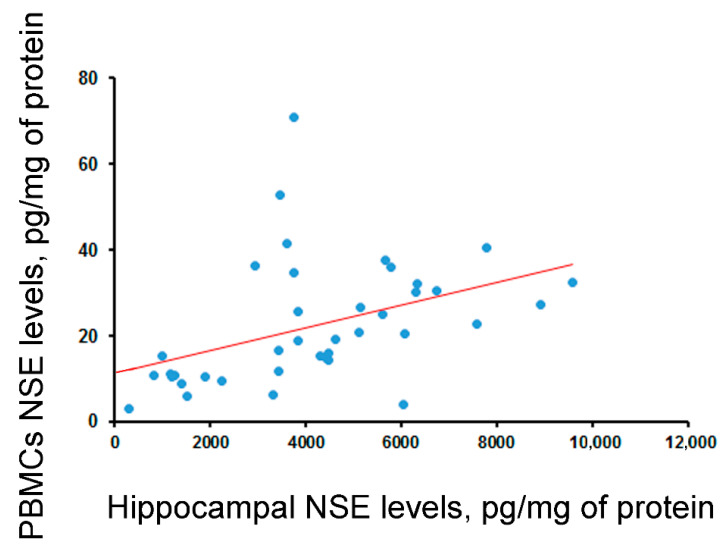
Correlation of concentration of NSE (corrected by total protein) in the hippocampuses and PBMCs in 40 operated epileptic patients (*rho* = 0.55, *p* < 0.001, Spearman test). The regression line is shown.

**Figure 4 ijms-25-00502-f004:**
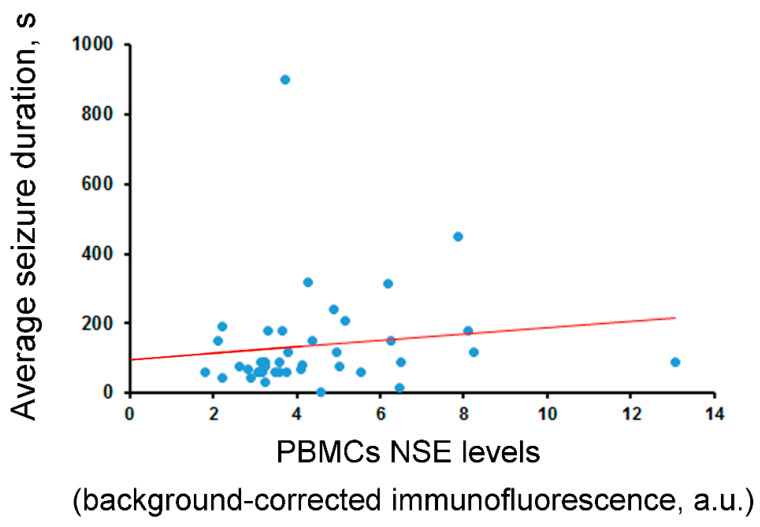
Correlation between NSE levels in PBMCs and average seizure duration in 41 operated epileptic patients (*rho* = 0.34, *p* < 0.05, Spearman test). The regression line is shown.

**Figure 5 ijms-25-00502-f005:**
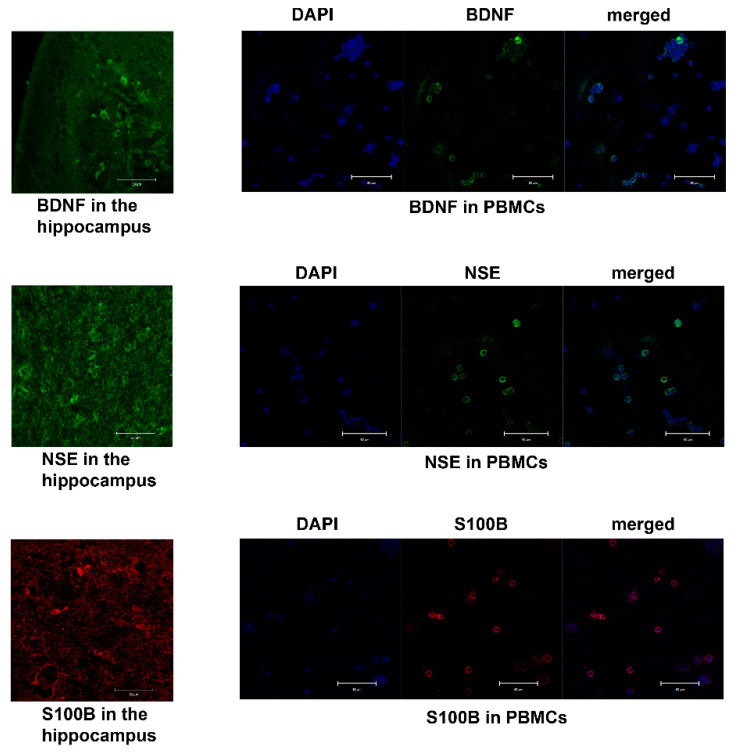
BDNF, NSE, and S100B immunoreactivity in the hippocampuses and PBMCs of patients with MTLE; scale bar: 50 μm.

**Table 1 ijms-25-00502-t001:** In vivo levels of BDNF, NSE, VILIP-1, and S100B in the hippocampus and blood of patients with epilepsy.

Analyte	Type of Analysis/Tissue				
	Biochemical Analysis			IHC Analysis	
	Hippocampus (pg/mg of Protein)	Blood		Hippocampus (a.u.)	Blood
		Serum (pg/mL)	PBMCs (pg/mg of Protein)		PBMCs (a.u.)
BDNF	224.8 (138.0; 361.1) (N = 38)	3458.7 (2707.6; 3891.9) (N = 41)	1243.2 (623.2; 2169.8) (N = 40)	0.027 (0.017; 0.037) (N = 43)	3.222 (2.14; 3.811) (N = 43)
NSE	4077.3 (2593.0; 5722.2) (N = 40)	1.8 (1.2; 2.8) (N = 40)	67.1 (33.1; 154.3) (N = 40)	0.0095 (0.0064; 0.0148) (N = 43)	3.749 (3.163; 5.163) (N = 42)
VILIP-1	98.6 (65.4; 190.4) (N = 39)	244.72 (155.1; 371.3) (N = 41)	124.9 (81.0; 228.9) (N = 39)	-	-
S100B	2298.3 (1800.4; 2942.5) (N = 39)	38.5 (23.5; 66.5) (N = 41)	1.55 (0.44; 4.85) (N = 39)	0.011 (0.008; 0.015) (N = 30)	1.842 (1.401; 2.274) (N = 30)

Data are presented as medians (Q1; Q3).

**Table 2 ijms-25-00502-t002:** Concordance of BDNF levels between the hippocampus and blood; some clinical correlates of brain or blood BDNF levels.

Parameter	Type of Analysis/Tissue				
	Biochemical Analysis			IHC Analysis	
	Hippocampus	Blood		Hippocampus	Blood
		Serum	PBMCs		PBMCs
Correlation between blood and hippocampal levels	-	*rho* = −0.11 (n.s., N = 38)	*rho* = −0.17 (n.s., N = 38)	-	*rho* = 0.15 (n.s., N = 43)
BDNF levels in female (f) vs. male (m) patients	f vs. m (*Z* = 1.08, n.s., N = 38)	f vs. m (*Z* = 1.22, n.s., N = 40)	f vs. m (*Z* = 1.12, n.s., N = 40)	f vs. m (*t* = 0.36, n.s., N = 43)	f vs. m (*Z* = 0.97, n.s., N = 43)
Correlation between BDNF levels and age (years)	*rho* = −0.18 (n.s., N = 38)	***r* = 0.32 (*p* = 0.045, N = 41)**	*rho* = 0.05 (n.s., N = 40)	*r* = −0.12 (n.s., N = 43)	*rho* = 0.13 (n.s., N = 43)
Correlation between BDNF levels and disease duration (years)	*rho* = −0.19 (n.s., N = 38)	*r* = 0.09 (n.s., N = 41)	*rho* = 0.09 (n.s., N = 40)	*r* = 0.17 (n.s., N = 43)	*rho* = −0.08 (n.s., N = 43)
Correlation between BDNF levels and average seizure frequency per month	*rho* = 0.15 (n.s., N = 38)	*rho* = 0.17 (n.s., N = 41)	***rho* = 0.46, *p* < 0.003 (N = 40)**	*rho* = −0.06 (n.s., N = 43)	*rho* = 0.18 (n.s., N = 43)
Correlation between BDNF levels and average seizure duration (s)	*rho* = −0.17 (n.s., N = 38)	*rho* = −0.003 (n.s., N = 41)	*rho* = 0.01 (n.s., N = 40)	*rho* = −0.06 (n.s., N = 43)	*rho* = 0.001 (n.s., N = 43)
BDNF levels in patients with hippocampal sclerosis (HS) vs. hippocampal gliosis (HG)	HS [0.23 ± 0.03 ng/mL] vs. HG [0.25 ± 0.01 ng/mL] (*F*(1, 36) < 1, N = 38)	HS [3.52 (2.80; 3.84) ng/mL] vs. HG [3.07 (2.53; 3.99) ng/mL] (*Z* = 0.73, *p* > 0.05, N = 41)	HS [1.48 (0.84; 2.49) ng/mL] vs. HG [1.03 (0.58; 1.59) ng/mL] (*Z* = 1.59, *p* = 0.11, N = 40)	HS [0.03 ± 0.003 a.u.] vs. HG [0.024 ± 0.003 a.u.] (*F*(1, 41) = 2.2, *p* = 0.15, N = 43)	HS [3.67 (2.79; 4.01) a.u.] vs. HG [2.3 (1.47; 3.33) a.u.] (***Z* = 2.87, *p* = 0.0041**, N = 43)
Outcome (ILAE classes)	*F*(5, 32) = 1.7, *p* > 0.05, N = 38	*F*(5, 35) < 1, N = 41	*H*(5, N = 40) = 9.53, *p =* 0.09	*F*(5, 37) < 1, N = 43	*H*(5, N = 43) = 6.12, *p* > 0.05
BDNF levels in patients who were treated with drugs stimulating GABAergic activity (1) vs. those who were not (0)	1 vs. 0 (*t* = −0.03, n.s., N = 38)	1 vs. 0 (*t* = 0.77, n.s., N = 41)	1 vs. 0 (*Z* = 1.00, n.s., N = 40)	1 vs. 0 (*t* = 0.2, n.s., N = 43)	1 vs. 0 (*Z* = −1.06, n.s., N = 43)
BDNF levels in patients who were treated with drugs regulating ion channel activity (1) vs. those who were not (0)	1 vs. 0 (*Z* = −0.21, n.s., N = 38)	1 vs. 0 (***t*** **= 2.70**, ***p* = 0.01**, N = 41)	1 vs. 0 (*Z* = 0.84, n.s., N = 40)	1 vs. 0 (*t* = −0.13, n.s., N = 43)	1 vs. 0 (*Z* = −0.37, n.s., N = 43)
BDNF levels in patients who were treated with drugs inhibiting the activity of NMDA receptors (1) vs. those who were not (0)	1 vs. 0 (*t* =−0.23, n.s., N = 38)	1 vs. 0 (*t* =−0.61, n.s., N = 41)	1 vs. 0 (*Z* = 0.35, n.s., N = 40)	1 vs. 0 (*t* = −0.05, n.s., N = 43)	1 vs. 0 (*Z* = −0.41, n.s., N = 43)

Data are presented as the medians (Q1; Q3) or means ± S.E.M.

**Table 3 ijms-25-00502-t003:** Concordance of NSE levels between the hippocampus and blood; some clinical correlates of brain or blood NSE levels.

Parameter	Type of Analysis/Tissue				
	Biochemical Analysis			IHC Analysis	
	Hippocampus	Blood		Hippocampus	Blood
		Serum	PBMCs		PBMCs
Correlation between blood and hippocampal levels	-	*rho* = −0.29 (*p* = 0.06, N = 40)	***rho* = 0.55 (*p* < 0.001, N = 40)**	-	*rho* = 0.3 (*p* = 0.057, N = 41)
NSE levels in female (f) vs. male (m) patients	f vs. m (*t* = −0.05, n.s., N = 40)	f vs. m (*Z* = −1.95, *p* = 0.05, N = 40)	f vs. m (*Z* = −0.14, n.s., N = 40)	f vs. m (*Z* = −1.68, n.s., N = 43)	f vs. m (*Z* = −0.68, n.s., N = 42)
Correlation between NSE levels and age (years)	*r* = −0.18 (n.s., N = 40)	*rho* = 0.10 (n.s., N = 40)	*rho* = −0.27 (n.s., N = 40)	*rho* = −0.08 (n.s., N = 43)	*rho* = 0.17 (n.s., N = 42)
Correlation between NSE levels and disease duration (years)	*rho* = −0.04 (n.s., N = 40)	*rho* = 0.05 (n.s., N = 40)	*rho* = −0.19 (n.s., N = 40)	*rho* = 0.04 (n.s., N = 43)	*rho* = −0.08 (n.s., N = 42)
Correlation between NSE levels and average seizure frequency per month	*rho* = 0.05 (n.s., N = 40)	*rho* = 0.03 (n.s., N = 40)	*rho* = 0.02 (n.s., N = 40)	*rho* = −0.09 (n.s., N = 43)	*rho* = 0.07 (n.s., N = 42)
Correlation between NSE levels and average seizure duration (s)	*rho* = −0.00 (n.s., N = 40)	*rho* = −0.06 (n.s., N = 40)	*rho* = −0.11 (n.s., N = 40)	*rho* = −0.09 (n.s., N = 42)	***rho* = 0.34** (***p* = 0.03**, N = 41)
NSE levels in patients with hippocampal sclerosis (HS) vs. hippocampal gliosis (HG)	HS vs. HG (*t* = −1.33, n.s., N = 40)	HS vs. HG (*Z* = 0.97, n.s., N = 40)	HS vs. HG (*Z* = −1.92, n.s., N = 40)	HS vs. HG (*Z* = −0.43, n.s., N = 43)	HS vs. HG (*Z* = 0.87, n.s., N = 42)
Outcome (ILAE classes)	*F*(5, 33) = 0.19, *p* > 0.05	*H*(5, N = 39) = 4.23, *p* > 0.05	*H*(5, N = 39) = 5.22, *p* > 0.05	*H*(5, N = 43) = 2.92, *p* > 0.05	*H*(5, N = 42) = 6.79, *p* > 0.05
NSE levels in patients who were treated with drugs stimulating GABAergic activity (1) vs. those who were not (0)	1 vs. 0 (*t* = 0.23, *p* = 0.056, N = 40)	1 vs. 0 (*Z* = −0.17, n.s., N = 40)	1 vs. 0 (*Z* = 1.67, n.s., N = 40)	1 vs. 0 (*Z* = 0.23, n.s., N = 43)	1 vs. 0 (*Z* = 0.52, n.s., N = 42)
NSE levels in patients who were treated with drugs regulating ion channel activity (1) vs. those who were not (0)	1 vs. 0 (*t* = −0.52, n.s., N = 40)	1 vs. 0 (*Z* = 1.00, n.s., N = 40)	1 vs. 0 (*Z* = 0.82, n.s., N = 40)	1 vs. 0 (*Z* = 1.5, n.s., N = 43)	1 vs. 0 (*Z* = 1.86, n.s., N = 42)
NSE levels in patients who were treated with drugs inhibiting the activity of NMDA receptors (1) vs. those who were not (0)	1 vs. 0 (***t* = 2.17**, ***p* < 0.05**, N = 40)	1 vs. 0 (***Z* = −2.52**, ***p* < 0.05**, N = 40)	1 vs. 0 (*Z* = 0.08, n.s., N = 40)	1 vs. 0 (*Z* = −1.65, n.s., N = 43)	1 vs. 0 (*Z* = −0.68, n.s., N = 42)

**Table 4 ijms-25-00502-t004:** Concordance of S100B levels between the hippocampus and blood; some clinical correlates of brain or blood S100B levels.

Parameter	Type of Analysis/Tissue				
	Biochemical Analysis			IHC Analysis	
	Hippocampus	Blood		Hippocampus	Blood
		Serum	PBMCs		PBMCs
Correlation between blood and hippocampal levels	-	*rho* = 0.14 (n.s., N = 39)	*rho* = 0.21 (n.s., N = 39)	-	*r* = 0.16 (n.s., N = 30)
S100B levels in female (f) vs. male (m) patients	f vs. m (*t* = 1.85, *p* = 0.07, N = 39)	f vs. m (*Z* = 0.86, n.s., N = 41)	f vs. m (*Z* = 0.43, n.s., N = 39)	f vs. m (*t* = −0.17, n.s., N = 30)	f vs. m (*t* = −0.45, n.s., N = 30)
Correlation between S100B levels and age (years)	*r* = −0.06 (n.s., N = 39)	***rho* = 0.34**, **(*p* < 0.05**, N = 41)	*rho* = 0.06 (n.s., N = 39)	*r* = −0.05 (n.s., N = 30)	*r* = −0.12 (n.s., N = 30)
Correlation between S100B levels and disease duration (years)	*r* = −0.23 (n.s., N = 39)	*rho* = 0.19 (n.s., N = 41)	*rho* = 0.10 (n.s., N = 39)	*r* = 0.16 (n.s., N = 30)	*r* = −0.22 (n.s., N = 30)
Correlation between S100B levels and average seizure frequency per month	*rho* = 0.26 (n.s., N = 39)	*rho* = 0.23 (n.s., N = 41)	*rho* = 0.27 (n.s., N = 39)	*rho* = −0.14 (n.s., N = 30)	*rho* = 0.004 (n.s., N = 30)
Correlation between S100B levels and average seizure duration (s)	*rho* = −0.05 (n.s., N = 39)	*rho* = 0.02 (n.s., N = 41)	*rho* = 0.05 (n.s., N = 39)	*rho* = −0.06 (n.s., N = 29)	*rho* = 0.28 (n.s., N = 29)
S100B levels in patients with hippocampal sclerosis (HS) vs. hippocampal gliosis (HG)	HS vs. HG (*t* = −0.20, n.s., N = 39)	HS vs. HG (*Z* = 1.39, n.s., N = 41)	HS vs. HG (*Z* = −0.12, n.s., N = 39)	HS vs. HG (*t* = −0.91, n.s., N = 30)	HS vs. HG (*t* = −1.31, n.s., N = 30)
Outcome (ILAE classes)	*F*(5, 33) = 0.30, *p* > 0.05, N = 39	*H*(5, N = 41) = 5.62, n.s.	*H*(5, N = 39) = 2.19, n.s.	*F*(5, 24) =1.04, *p* > 0.05	*F*(5, 24) = 1.46, *p* > 0.05
S100B levels in patients who were treated with drugs stimulating GABAergic activity (1) vs. those who were not (0)	1 vs. 0 (*t* = −1.08, n.s., N = 39)	1 vs. 0 (*Z* = 0.64, n.s., N = 41)	1 vs. 0 (*Z* = −0.09, n.s., N = 39)	1 vs. 0 (*t* = 1.58, n.s., N = 30)	1 vs. 0 (*t* = 0.43, n.s., N = 30)
S100B levels in patients who were treated with drugs regulating ion channel activity (1) vs. those who were not (0)	1 vs. 0 (*t* = −0.92, n.s., N = 39)	1 vs. 0 (*Z* = 0.85, n.s., N = 41)	1 vs. 0 (*Z* = −0.76, n.s., N = 39)	1 vs. 0 (*t* = −0.42, n.s., N = 30)	1 vs. 0 (*t* = 0.42, n.s., N = 42)
S100B levels in patients who were treated with drugs inhibiting the activity of NMDA receptors (1) vs. those who were not (0)	1 vs. 0 (*t* = 1.44, n.s., N = 39)	1 vs. 0 (*Z* = −0.12, n.s., N = 41)	1 vs. 0 (*Z* = 0.08, n.s., N = 39)	1 vs. 0 (*t* = 1.61, n.s., N = 30)	1 vs. 0 (*t* = −0.66, n.s., N = 30)

**Table 5 ijms-25-00502-t005:** Concordance of VILIP-1 levels between the hippocampus and blood; some clinical correlates of brain or blood VILIP-1 levels.

	Biochemical Analysis		
Parameter	Tissue		
	Hippocampus	Blood	
		Serum	PBMCs
Correlation between blood and hippocampal levels	-	*rho* = 0.11 (n.s., N = 39)	*rho* = 0.01 (n.s., N = 39)
VILIP-1 levels in female (f) vs. male (m) patients	f vs. m (*Z* = 1.93, *p* = 0.054, N = 39)	f vs. m (*Z* = −1.03, n.s., N = 41)	f vs. m (*Z* = −0.49, n.s., N = 39)
Correlation between VILIP-1 levels and age (years)	*rho* = 0.09 (n.s., N = 39)	*rho* = 0.10 (n.s., N = 41)	*rho* = −0.26 (n.s., N = 39)
Correlation between VILIP-1 levels and disease duration (years)	*rho* = 0.05 (n.s., N = 39)	*rho* = 0.01 (n.s., N = 41)	*rho* = −0.07 (n.s., N = 39)
Correlation between VILIP-1 levels and average seizure frequency per month	*rho* = 0.24 (n.s., N = 39)	*rho* = 0.10 (n.s., N = 41)	*rho* = 0.10 (n.s., N = 39)
Correlation between VILIP-1 levels and average seizure duration (s)	*rho* = −0.03 (n.s., N = 39)	*rho* = −0.05 (n.s., N = 41)	*rho* = −0.23 (n.s., N = 39)
VILIP-1 levels in patients with hippocampal sclerosis (HS) vs. hippocampal gliosis (HG)	HS vs. HG (*Z* = 1.93, n.s., N = 39)	HS vs. HG (*Z* = 0.95, n.s., N = 41)	HS vs. HG (*Z* = −1.13, n.s., N = 39)
Outcome (ILAE classes)	*H*(5, N = 39) = 3.27, n.s.	*H*(5, N = 41) = 1.13, n.s.	*H*(5, N = 39) = 4.38, n.s.
VILIP-1 levels in patients who were treated with drugs stimulating GABAergic activity (1) vs. those who were not (0)	1 vs. 0 (*Z* = −0.25, n.s., N = 39)	1 vs. 0 (*Z* = −0.74, n.s., N = 41)	1 vs. 0 (*Z* = 0.64, n.s., N = 39)
VILIP-1 levels in patients who were treated with drugs regulating ion channel activity (1) vs. those who were not (0)	1 vs. 0 (Z = −0.98, n.s., N = 39)	1 vs. 0 (Z = −0.30, n.s., N = 41)	1 vs. 0 (Z = −0.71, n.s., N = 39)
VILIP-1 levels in patients who were treated with drugs inhibiting the activity of NMDA receptors (1) vs. those who were not (0)	1 vs. 0 (Z = 1.45, n.s., N = 39)	1 vs. 0 (Z = −1.88, *p* = 0.06, N = 41)	1 vs. 0 (Z = 0.39, n.s., N = 39)

## Data Availability

The data presented in this study are available upon request from the corresponding author.

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
