# Peer review of "Concordance between the In Vivo Content of Neurospecific Proteins (BDNF, NSE, VILIP-1, S100B) in the Hippocampus and Blood in Patients with Epilepsy"

_ijms, 2023, doi:10.3390/ijms25010502_

Round 1
Reviewer 1 Report
Comments and Suggestions for Authors
Comments attached

Author Response
We would like to thank the Reviewer for his/her valuable comments and suggestions and wish the best season’s greetings. We have thoroughly revised the manuscript considering all the comments, which helped us to improve the manuscript. All major corrections made in the text are marked with blue color. We believe that the revised version would be more clear and interesting for the readership of the journal.
- The data presentation of all four parameters in the form of tabulations is confusing and should be more clear presentation in graphical presentation.
We analyzed four neurospecific substances in hippocampus, serum, and leukocytes. Moreover, the levels of BDNF, NSE, and S100B were analyzed in the hippocampus and leukocytes both by biochemical methods and immunohistochemical analysis. In addition, we analyzed correlations and associations with a number of clinical parameters. We suggest that such a big number of parameters would be more comprehensive in a form of tables. Nevertheless, we added four figures (Figure 1-4) of the most significant results to illustrate and emphasize them.
- The author needs to provide the data of serum an IHC data in separate figures/graphical presentation with their correlation status.
Sorry, we did not get your idea. It is not possible to measure serum levels of the analytes using IHC analysis. We did not compare data obtained by biochemical and IHC methods. All correlations analyzed for serum or data obtained by IHC are presented in the tables and Figures 1-4.
- The author should provide the patient population parameter of sample size in separate table including sex, age, and stage of the disease.
The detailed information on patients is presented in Supplementary materials, Table S1.
- The author needs to establish the correlation between these parameters with the degree of disease.
We have analyzed correlations between the levels of neurospecific substances and some clinical parameters related to the degree of disease including disease duration, average seizure frequency per month, average seizure duration, ILAE class.
- The authors presented these for parameters in blood and hippocampus further Please clarify the correlation of these parameters in regard to CSF and other biological fluids.
Among human peripheral tissues and fluids, the blood attracts particular attention because it is easily accessible for research and wide screening in population (vs. CSF or nasal epithelium). Moreover, unlike blood, CSF is considered to be poorly suitable for the determination of BDNF, presumably due to the low analyte content and invasiveness of the method [Carlino et al., 2011, doi: 10.1016/j.jpsychires.2010.06.012]. On the other hand, the concentration of neurospecific proteins in the blood serum is considered to be closer to brain content than in the urine or saliva. Thus, in this study we focus on correlations between central levels in the brain (hippocampus) and blood. We did not analyze correlations in regard to CSF or other biological fluids.
Reviewer 2 Report
Comments and Suggestions for Authors
The article entitled: Concordance between the in vivo content of neurospecific proteins (BDNF, NSE, VILIP-1, S100B) in the hippocampus and blood in patients with epilepsy by Tikhonova et al., is well designed and written. The methodologies are appropriate and aligned with the proposed objectives. The images used are very suggestive and of a superior quality. The conclusions are consistent with the evidence and arguments presented. Very well-chosen statistical analysis methods. All references used by the authors are appropriate. The limitations of the present study are well presented. The study seems interesting, it could be accepted due to significance and further advantages compared to the existing literature.
Author Response
We would like to cordially thank the Reviewer for the careful review of our manuscript and wish the best season’s greetings. We greatly appreciate his/her high esteem of our study.
Reviewer 3 Report
Comments and Suggestions for Authors
Dear colleagues,
do not consider the works offered to you for review as arrogance and for imposing citations.
The authors of these papers tried to link the severity and prognosis with the neuromarkers in the blood serum of children with traumatic brain injury that are close to you.
- Polymorphism of the APOE Gene and Markers of Brain Damage in the Outcomes of Severe Traumatic Brain Injury in Children. Neuroscience and Behavioral Physiology, Vol. 51, No. 1, January, 2021 DOI 10.1007/s11055-020-01035-5
- Brain Biomarkers in Children After Mild and Severe Traumatic Brain Injury. Acta Neurochir Suppl. 2021;131:103-107.doi: 10.1007/978-3-030-59436-7_22. PMID: 33839828.
- Evolutionary aspects of a typical pathological process. Open J Clin Med Case Rep. 2023; Volume 9 (2023) Issue 17 2042. ISSN: 2379-1039 IF=2,1 https://doi.org/10.52768/2379-1039/2042

Author Response
We would like to cordially thank the Reviewer for the careful review of our manuscript and wish the best season’s greetings. We greatly appreciate her high esteem of our study. We have revised the manuscript considering the suggested references (#36 and #50), which helped us to improve the manuscript. All major corrections made in the text are marked with blue color. We believe that the revised version would be more clear and interesting for the readership of the journal.